# Influence of Polymer Modifiers on Selected Properties and Microstructure of Cement Waterproofing Mortars

**DOI:** 10.3390/ma14247558

**Published:** 2021-12-09

**Authors:** Wacław Brachaczek, Adam Chleboś, Zbigniew Giergiczny

**Affiliations:** 1Faculty of Materials, Civil and Environmental Engineering, University of Bielsko-Biala, ul. Willowa 2, 43-309 Bielsko-Biala, Poland; 2Faculty of Civil Engineering, Silesian University of Technology, ul. Akademicka 5, 44-100 Gliwice, Poland

**Keywords:** polymer-modified mortar, adhesion, admixtures, polymer modifiers, microstructure

## Abstract

This paper presents the results of research on the influence of polymer modifiers: styrene-acrylic copolymer, vinyl acetate/ethylene (EVA), vinyl acetate/acrylic copolymer (VAAc), and VA/VeoVa/acrylic terpolymer on the water permeability and adhesion of cement-containing waterproofing mortars in concrete. The content of the polymers in the composition of the mortars was 15, 20 and 26% (m/m) in relation to the weight of the dry ingredients. Using microscopic methods, an attempt was made to analyse the relationship between the microstructure of the mortars and the properties of these polymers. The EVA and the vinyl acetate/acrylic copolymer, which were used in the form of dry powders, had the most favourable effect on water permeability and adhesion to the concrete substrate. They may prove to be useful for the production of one-component cement-containing waterproofing mortars. On the other hand, the VA/VeoVa/acrylic terpolymer modifier had the least favourable effect on the tested properties. For mortars with this modifier, the desired water-permeability parameters were not achieved. Depending on the amount of polymer modifier, the mortars were characterized by differences in watertightness, as established on the basis of changes in porosity and differences in the adhesion of the cement-polymer paste to the surface of aggregate grains. It was determined that the type of polymer and its dispergation properties influence the water permeability of mortars, as well as their adhesion to concrete substrates.

## Highlights

Effects of different polymer latexes on watertightness and adhesion.Microstructures of polymer-modified mortars.

## 1. Introduction

Waterproofing mortars are used to protect concrete structures against the effects of water. They are used in systems that are applied to secure concrete structures embedded in the ground, and as composite waterproofing for the sealing of swimming-pool basins, both new and renovated. The suitability of the components for making waterproofing mortars is determined by technological features such as the ease of preparation and application, and functional features, including tensile and bending strength, watertightness, adhesion to the surface of various materials and resistance to environmental factors. The addition of a polymer affects the properties of mortars in both plastic and hardened forms. Polymers improve flexibility, crack-bridging properties and adhesion to the substrate. The microstructure changes are influenced by the simultaneous cement-hydration reactions and the polymer-coalescence process [1,2,3,4,5,6].

The speed of the setting and hardening processes of waterproofing mortars is varied and cement hydration is usually preceded by the phenomenon of polymer-film formation. With the loss of water in the mortar, as a result of cement hydration, the polymer-coalescence process begins. The polymer phase combines with the crystalline network of cement-hydration products, resulting in a continuous “co-matrix phase”, in which the cement-hydration products and the polymer phase interpenetrate. The matrix that is formed in this way can be influenced by the type of polymer modifier, its amount, and the form (liquid, loose) in which it is introduced into the composition of the mortar [2,3,4,7,8]. 

Products based on homopolymers, copolymers and tripolymers are used to modify cement mortars. The most important are polyvinyl acetate homopolymers and polyacrylates. The most common among the copolymers are the following: styrene-acrylic and vinyl acetate/ethylene copolymers, and polyvinyl acetate with acrylates. 

Among the relatively new modifiers are copolymers from the vinyl acetate group and from vinyl ester of the edetic acid (VeoVa) group [9,10,11]. The VeoVa structural unit embedded in the polymer chain increases resistance to UV radiation, lowers surface tension and increases water-repellent properties. 

The properties of polymers are also influenced by other factors, such as the mutual proportions of individual structural units in the polymer chain, the degree of chain branching and the length of the polymer chains [12]. Polymers differ in terms of viscosity and the minimum temperature at which the polymer particles combine to form a flexible and continuous coating. This temperature is referred to as the MFFT (the minimum film-forming temperature) [13,14]. 

Polymer modifiers can be found both in the form of aqueous dispersions and redispersible powders [1,15,16,17,18]. Waterproofing mortars with polymer modifiers are offered as one-component products (mixtures of dry components) or two-component products (dry components of the mortar plus a liquid polymer). The influence of the polymer form on the properties of hardened cement composites has been the subject of research [16,19]. The author of [17] used polymers in the form of aqueous dispersions rather than in the form of powders in order to obtain better effects of the modification of the cement materials. The authors of [16] also demonstrated that the aggregate state of the polymer modifiers influences the uniform distribution of the polymer in the hardened mortar. Based on microscopic SEM observations, they noticed significant differences in the structure of the film that was produced. They found that the microstructure of the mortars that were modified by the use of powders was heterogeneous, fibrous, and porous with numerous thickenings and unevenness.

The selection of the polymer, both qualitative and quantitative, is based on the designer’s knowledge and previous practical experience. Various design concepts, which consist of examining the relationship between the properties of the modified mortars and the porosity, the w/c ratio or the cement content, prove to be helpful in this respect. Drawing an analogy with the repair materials used to protect and repair concrete structures, it can be assumed that the adhesion of the waterproofing should not be lower than the tensile strength of the concrete sub-base [20]. This means that the waterproofing of high-strength concrete will require the use of mortars characterized by higher adhesion. Therefore, it is advisable to conduct systematic research explaining the influence of polymer modifiers on the properties of waterproofing mortars after hardening. 

This study analyses the effect of the amount and type of polymer on the principal functional properties of polymer-modified mortars, i.e., adhesion and water permeability (leak-tightness). Based on the microscopic observations, the influence of the type of polymer modifier on the microstructure of the hardened mortar was assessed. An attempt was also made to explain the relationship between the microstructure and the adhesion and watertightness. 

## 2. Experimental Material and Testing Method

### 2.1. Experimental Material

Four industrially produced polymer modifiers were selected for the study: **ZH1/ASA**—Styrene-acrylic copolymer—in the form of an aqueous dispersion with a solids content of 55–57%, a grain size of 200–400 nm, an MFFT temperature of 1 °C, and a styrene:acrylic ratio of 6:4.**ZH2/CF/EVA**—EVA ethylene-vinyl acetate copolymer—redispersible powder with an average particle size of 90 µm and an MFFT temperature of 0 °C, with 20% of ethylene.**ZH3/ORG/VAAc**—VAAc vinyl acetate-acrylic copolymer—hydrophobic redispersible powder with an MFFT temperature of 0 °C.**ZH4/VA/VeoVa**—vinyl acetate-vinyl versatate-acrylic VA/VeoVa/acrylic terpolymer—redispersible powder with an MFFT temperature of 0 °C.

The influence of the above-mentioned polymer modifiers on the properties of waterproofing mortars was examined using CEM I 42.5 R Portland cement according to PN-EN 197-1:2012 [21]. The properties of the cement are presented in Table 1 and Table 2. The mortars also contain CL 90-S hydrated lime according to PN-EN 459-1:2015-06 [22]. The aggregates that were used are: quartz sand with a grain size of 0.0–0.5 mm, sand with a grain size of 0.1–0.3 mm and quartz powder with a grain size of 0.0–0.1 mm. The size distribution of the aggregates is presented in Figure 1. In order to give the mortars the appropriate consistency, flow, and to reduce runoff from vertical surfaces and ensure water retention, as well as to eliminate air voids from the structure, a cellulose thickener—hydroxyethyl methylcellulose with a viscosity of 25,000–35,000 cps (2% solution tested at 20 °C using a Brookfield viscosity gauge)—was used together with starch ether, which had a viscosity of 15 mPas (2% solution tested at 20 °C using a Höpler viscosity gauge). In order to increase the tensile strength and improve flexibility, FPE polyethylene fibres with a length of 200–500 µm and a diameter of approx. 10 µm, containing a PVA-wetting agent (approx. 1% m/m), and cellulose fibres with a length of up to 400 µm and a diameter of up to 45 µm were added. The addition of fibres also allows for the creation of a spatial skeleton, allowing the mortar to seal small scratches in the substrate. 

The studies were carried out on a test mortar recipe that was developed using Ohama’s method [23,24]. The amounts of cement, aggregates and admixtures as well as the polymer-to-cement ratio were determined on the basis of the desired adhesion to the concrete substrate of 1.0 MPa. The composition of the test mortar is provided in Table 3. 

The polymer modifier was added in amounts of 15, 20 and 26% m/m. The dosing of 26% of the modifier was caused by technological reasons. The differences in weight resulting from the change in the amounts of modifiers were compensated with 0.0–0.5 mm grain-size sand.

### 2.2. Experimental Procedure

The scope of our own experimental research included the assessment of the water permeability and adhesion of the tested insulation mortars as well as the assessment of their microstructure after hardening. 

Water permeability was determined using mortar samples applied to cubic concrete blocks with sides of 15 cm. The mortars were applied in two layers, each 2 mm thick, to one of the walls. The remaining walls of the cubes were sealed with acrylic-resin-based material. The samples were stored for 28 days at 22 °C and 50% relative air humidity. After this period, the samples were dried at 22 °C to a constant weight and placed in a device for testing water permeability in accordance with the procedure specified in PN-EN 14891:2017-03 [25]. The mortar samples were exposed to water under a pressure of 0.15 MPa for 7 days. After completion of the test, the cubes with mortar were immediately weighed. The measure of water permeability was the amount of absorbed water. Mortars were considered tight if the amount of absorbed water did not exceed 20 g. The reference point for the assessment of water permeability were the mortars without polymer modifiers.

The adhesion of the mortars to the substrate was determined using the pull-off method. The initial adhesion was tested 28 days after application, in accordance with the procedure specified in PN-EN 14891:2017-03 [25], in two 2 mm-thick layers, each on a concrete substrate of compressive-strength class C12/15. The adhesion of the sealing mortars when performing composite waterproofing should not be less than 1 MPa. The reference point for the assessment of adhesion were the mortars without polymer modifiers.

The impact of polymer modifiers on the microstructure of hardened mortars was determined using the SEM Phenom ProX (Phenom World) electron-microscopy technique with the EDS (Energy Dispersive Spectroscopy) system operating at a voltage of 10 kV. The observations were carried out after the samples were coated with a thin layer of gold (sputter deposition method) and at a magnification of 2250×. 

## 3. Test Results and Analyses

### 3.1. Water Permeability

The results of the water-permeability tests of waterproofing mortars are presented in Figure 2. 

The mortar without a polymer modifier was characterized by a high water permeability of 190 g. The mortars with the ZH2/CF/EVA modifier were characterized by the highest watertightness. In this case, already at 15% m/m, the amount of water absorbed was 20 g. A further increase in the amount of the modifier had no significant effect on water permeability. For the mortars modified with ZH3/ORG/VAAc in the amount of 15% m/m, 72 g of water was absorbed, which is nearly 2.5 times more than in the case of the mortar containing the same amount of the ZH2/CF/EVA modifier. In this case, the desired mortar watertightness was obtained using 20% m/m of the polymer. 

After analysing the results of the water-permeability tests with the ZH4/VA/VeoVa modifier in the VeoVa groups, it can be concluded that its effectiveness is lower as compared to the ZH2/CF/EVA polymer. In this case, with a polymer content of 15% m/m the water absorption was 45 g, and at 20% m/m it decreased to 30 g. A further increase in the amount of this modifier had little effect on the watertightness. In summary, it should be concluded that the application of this modifier did not produce a mortar that could be classified as watertight. 

Clearly different results were obtained for the ZH1/ASA polymer modifier that was used in the form of an aqueous dispersion. It can be seen (Figure 2) that its positive effect on watertightness is evident only at 26% m/m of the polymer. In view of the above, the influence of the amount of cement on the water permeability of the mortar with the ZH1/ASA modifier was analysed. The results are presented in Figure 3. No clear influence of the amount of cement on the amount of absorbed water was observed. 

### 3.2. Adhesion of the Mortar to Concrete Substrate

One of the most important features of waterproofing mortars is their adhesion to various materials, mainly concrete matrices. The mortar without the polymer modifier was characterized by a low adhesion to the concrete matrix of only 0.4 MPa. The results of adhesion measurements for mortars, which differed in type and amount of polymer modifier, to the concrete substrate are presented in Figure 4. After analysing the results, it can be concluded that the addition of the polymer in an amount of less than 15% m/m had little effect on the adhesion. A clear increase in adhesion was observed only with the addition of a polymer modifier in the amount of 20% m/m. In the case of ZH1/ASA, ZH2/CF/EVA and ZH3/ORG/VAAc, the limit amount of the modifier, at which point the adhesion is the highest, was determined to be 20% m/m. Once this amount is exceeded, the adhesion begins to decrease again. 

The ZH3/ORG/VAAc and ZH1/ASA modifiers had the greatest positive impact on the improvement of mortar adhesion. At a 20% m/m content of these modifiers, the adhesion significantly exceeded 1 MPa. In the case of the ZH2/CF/EVA polymer modifier the adhesion was close to 1 MPa. Increasing the proportion of ZH1/ASA, ZH2/CF/EVA and ZH3/ORG/VAAc modifiers to 26% m/m resulted in a reduction in adhesion. 

The effect of the ZH4/VA/VeoVa polymer was different. With the increase in the amount of ZH4/VA/VeoVa, the adhesion of the mortar increased uniformly to 0.55 MPa and it was much lower at 20% m/m than for mortars containing the other modifiers. At 26% m/m, the adhesion only reached 0.6 MPa (Figure 4). 

The effect of the simultaneous change in the amount of modifier and cement (% m/m) on the adhesion of the mortar (MPa) is presented in the example of the waterproofing mortar containing the ZH1/ASA polymer modifier. The results are presented in Figure 5. The graph has a clear inflection at the point where the modifier amount is 20% m/m. At 33% m/m of cement, the adhesion of the mortar is 1.2 MPa. It can also be stated that the content of the polymer modifier in the composition of the mortar has a greater influence on the adhesion than the content of the cement. A slight change in adhesion (MPa) of approximately 0.2 MPa was observed for a wide range of changes in the amount of cement, from 15 to 33% m/m, at constant amounts of polymer, while the adhesion changed by 0.8 MPa when the amount of modifier was changed from 11% m/m to 20% m/m. A well-known relationship is also the fact that too much cement increases the shrinkage of the mortar. This is disadvantageous in terms of both adhesion and watertightness.

### 3.3. The Microstructure of Mortars

The microstructure of the mortars was analysed, mainly in terms of polymer distribution, in accordance with the methodology presented in Section 2.2, in order to determine the homogeneity of the sample. The distribution of polymer and hydrates in the mortar was investigated by the microanalysis of the chemical composition of the samples’ fracture surfaces with the use of EDS. The test samples were taken from the surface, from the inner part of the mortar and from the mortar–substrate interface. The distribution of Si, C, Ca and Al atoms was analysed. The presence of carbon in the mortar was linked to the distribution of the polymer modifier of an organic nature (Figure 6). The tests were performed immediately after the completion of the seasoning of the samples in order to eliminate the risk of carbonation. Samples were cut using a low-speed saw and were not polished, as this could lead to the accelerated carbonation of the cement-hydration products, which we wanted to avoid. Additionally, the samples were kept in plastic bags until the start of the test. 

The microstructures of the mortars that were modified with ZH1/ASA, ZH2/CF/EVA and ZH3/ORG/VAAc were similar. Figure 6a shows the surface view of the mortar containing the ZH1/ASA polymer modifier in the amount of 15% m/m. Figure 6b shows the view of the mortar surface and the places where the presence of the polymer modifier was identified. In Figure 7, the surface views of the samples that were modified with the ZH3/ORG/VAAc in the amounts of 15% m/m (Figure 7a) and 20% m/m (Figure 7b) are compared. The samples were taken from the outer surface on which the areas of polymer-modifier occurrence were marked. After analysing the mapping results, it was found that the inter-grain space was filled with a cement-polymer paste with a homogeneous microstructure. Apart from the polymer, the paste featured evenly distributed hydrates and fine aggregates. Similar results were obtained for the samples taken from the inner parts of the mortar samples.

Figure 8 presents the fractures of the samples taken from the inner part of the mortar that were modified with ZH1/ASA in the amount of 20% m/m. The pores visible in Figure 8 can be divided into slotted pores and capillary pores. The slotted pores were mainly present in the paste–aggregate interfaces, while the capillary pores were visible in the inter-grain space of the paste and their diameters ranged from a few to over a dozen micrometres. It was found that with a smaller amount of polymer modifier in the mortar (less than 15% m/m) these pores were interconnected, creating a network that allowed the transport of solutions. Similar microstructures were observed for mortars containing the ZH2/CF/EVA and ZH3/ORG/VAAc polymer modifiers. It was found that the increase in the amount of modifiers in the mortar caused the pastes to adhere to the aggregates to a greater extent. The diameter of the pores and their number also decreased, which explains the reduction in the water permeability of the hardened mortars. 

The highest homogeneity of the microstructure and the lowest porosity were observed in the mortars with 26% m/m of the modifier (Figure 9). After analysing the changes in the microstructure of the mortars in the direction from the substrate towards the outer surface of the samples, it was found that the outer surface was richer in polymer than the samples from its middle part. Figure 9a shows an SEM photo of the surface of the mortar that was modified with ZH2/CF/EVA in the amount of 26% m/m. The microstructure of the mortar is homogeneous and tight. The small pores with diameters of a few micrometres can mainly be related to the evaporation of water during drying. Slotted pores appeared spontaneously in different places in the mortar. Their presence can be explained by the generation of stresses during the hardening of the mortar. Larger amounts of polymer were also observed in samples taken from the mortar–substrate interface (Figure 9b). This may indicate that during drying, the modifier and the paste were transported to the surface of the interphases. This phenomenon can be attributed to the influence of capillary forces and evaporation, which induce streams of solutions to flow in the interconnected pore system [26]. 

Mortars containing the ZH4/VA/VeoVa polymer modifier were characterized by a heterogeneous microstructure. Figure 10a shows the surface of the hardened mortar with the ZH4/VA/VeoVa modifier in the amount of 20% m/m, whereas Figure 10b presents the view of the fracture surface from the inner part of the mortar with the same amount of the modifier. It was found that the inter-grain space was filled with poorly adhering, heterogeneous polymer–cement paste with numerous pores of a few to over a dozen micrometres in diameter. On the outer surface of the samples, as well as on the surface of the fractures, it was possible to distinguish areas where the polymer formed a uniform film without the inclusion of hydrates or other inorganic substances (area 1, Figure 10a). In other places, clusters of hydrates together with cement grains of several dozen micrometres in diameter were visible (area 2, Figure 10b). The hydrates took either a fibrous shape, constituting the C-S-H phase with embedded ettringite decomposition products, or in other places irregularly shaped hydrates with a large number of isometric and flattened particles were visible (Figure 10b). Deep slot-shaped pores were visible in the aggregate–paste interfaces. There were also cylindrical pores in the intergranular space of the paste. Lower porosity of the mortar was observed in the mortar–substrate-interface plane and on the surface, which could be explained by the transport of polymer and cement with the water stream in these directions during drying. 

The further increase in the amount of polymer to 26% m/m favoured the reduction in porosity. In the microstructure of the mortar, places where a continuous, homogeneous polymer film dominates can be observed. The formed hydrates together with single cement grains appeared next to the polymer in the form of clusters (Figure 11b). The pores visible in the microstructure were interrupted by hardened polymer, which reduced the patency of these capillaries and reduced water permeability. Figure 11a shows the fracture of the mortar taken from the inner part of the mortar sample that was modified with ZH4/VA/VeoVa in the amount of 20% m/m, showing the pores with a circular cross-section filled with hardened polymer (area 1 and area 2). 

## 4. Discussion

This paper describes the influence of polymer modifiers on the principal properties of waterproofing mortars, i.e., water permeability and adhesion to the concrete substrate. 

After assessing the water permeability of the tested mortars, it can be concluded that it coincides with the degree and dispersibility of the polymers. The best results were obtained when the added polymers resulted in a uniform polymer–cement paste after hardening. The lowest water permeability was found for the mortar that was modified by the addition of the ZH2/CF/EVA polymer, which was an ethylene-vinyl acetate copolymer in powder form. The water-impermeable mortar was obtained already at 15% m/m of the modifier. A further increase in the amount of the modifier had no significant effect on the water permeability. The grains of this polymer in the mortar were well miscible with the mortar components and did not tend to form larger agglomerates (Figure 9a,b). Similar properties characterized the mortars to which the ZH1/ASA polymer modifier was added. This modifier was a styrene-acrylic copolymer added in the form of an aqueous dispersion and consisted of uniformly sized particles suspended in water, which formed during the polymerization stage. The polymer particles were evenly dispersed throughout the hardened mortar and showed good adhesion to the aggregates. The increase in the watertightness of the mortar with the greater amount of polymer can be associated with the gradual filling of the micro-, meso- and macropores by the polymer. 

After analysing the results of the studies on the effect of the amount of cement on water permeability, it can be concluded that water permeability depends to a greater extent on the proportion of polymer than of cement. The influence of the amount of cement on this property was ambiguous and requires further research.

The analysis of the influence of polymer modifiers on adhesion leads to the conclusion that the influence is diversified. In the case of the ZH1/ASA, ZH2/CF/EVA and ZH3/ORG/VAAc modifiers, the effect of amount on adhesion was similar. Modification of mortars with these polymers in the amount of 15% m/m was not sufficient. The highest adhesion was observed with 20% m/m of these polymers. A further increase in the amount of polymers reduced the adhesion. The low adhesion to the substrate of mortars containing 15% m/m of the polymer is explained by their microstructure. With this amount, it was porous (Figure 6a and Figure 7a), and the paste adhered poorly to the surface of the aggregate grains. Hydrates did not form a mineral skeleton, but rather they were present in agglomerates, depending on the type of polymer. At 20% m/m of polymer in the mortar, the hydrates were smaller than those at 15% m/m. On the other hand, a greater amount of dispersed hydrates was observed in the polymer matrix. After hardening, the larger aggregates were covered with a thin polymer film which was penetrated by diffused hydrates, thereby reducing the porosity of such a composite (Figure 10a). The amount of 20% m/m of the polymer in the mortar appears to be optimal in these systems because it allowed for the formation of fine hydrates, which favoured the development of their specific surface. This is advantageous because the expansion of the surface of the hydrates allows for a better use of Van der Waals’ attractive forces in the paste–aggregate interface (contact zone) [27,28]. The polymer, on the other hand, filled the pores of the formed skeleton well, and its greater amount that accumulated on the interphase surface supported the adhesion of the mortar. With the polymer amount of 26% m/m the influence of hydrates on adhesion is lower. The polymer accumulated in the plane of the paste–substrate interface may play a greater role. The adhesion decreased for mortars in which the proportion of polymer predominated over the amount of cement.

Among the tested polymer modifiers, the ZH4/VA/VeoVa polymer modifier was characterized by the lowest effectiveness, both in terms of water permeability and adhesion. Based on microscopic observations it can be concluded that it was less dispersible in the mixing water than the other modifiers. In fresh mortar it did not form a homogeneous dispersion with the cement grains. In hardened mortars, it was distributed unevenly, with a tendency to form clusters resembling plugs stuck in the inter-grain spaces, next to hydrates that were forming agglomerates of various sizes (Figure 11b). The observed microstructure can be associated with high water permeability, which also occurred at 26% m/m of this polymer in the mortar. Cracks and fissures, which made it permeable to water, were visible in the hardened paste. 

The obtained results prove that the microstructure of waterproofing mortars is diverse and depends on the type of polymer modifier that is used and its amount. Changes in the microstructure can be explained by two parallel mechanisms occurring during hardening: cement hydration and polymer coalescence. In the studied mortars, modifiers in the form of liquid dispersions as well as dry powders were used. Dispersion in the liquid paste of the polymer powders is possible thanks to the use of surfactants in the form of dispersants, emulsifiers, etc. It was found that it is not the form of the polymer modifier, but its chemical composition and properties, and in particular its ability to disperse with water in the liquid paste, that plays a major role in shaping the microstructure. These observations may seem different from those presented in the works by other researchers, such as All [16] and Adler [17]. 

On the basis of the obtained results, it was found that modifiers in the form of dry powders easily disperse into the paste. This demonstrates significant progress in the manufacture of redispersible polymers that are currently available, which is of great importance in terms of the technological aspect. They can be used to produce single-component waterproofing mortars with the desired performance characteristics.

## 5. Conclusions

Both water permeability as well as the adhesion of waterproofing mortars to substrates largely depend on the tendency of a polymer to disperse with cement and aggregates in the fresh mortar.The greatest impact on the water permeability of mortars was by the ZH2/CF/EVA modifier, i.e., the EVA-based ethylene-vinyl acetate copolymer. In fresh mortars it was easily dispersed, and after hardening it formed a homogeneous cement–polymer paste with evenly distributed hydrates and finer aggregates, which tightly filled the inter-grain space of the mortar.No significant effect on the adhesion and tightness of the mortars was noted after a change in the amount of cement. This problem, however, requires more research in the future.The powder form of the polymer modifiers was not a contra-indication to use. Polymer modifiers that were available on the market in the form of dry powders were characterized by an appropriate degree of dispersion in the mortar, which enabled the obtention of waterproofing mortars with adequate adhesion and watertightness.

## Figures and Tables

**Figure 1 materials-14-07558-f001:**
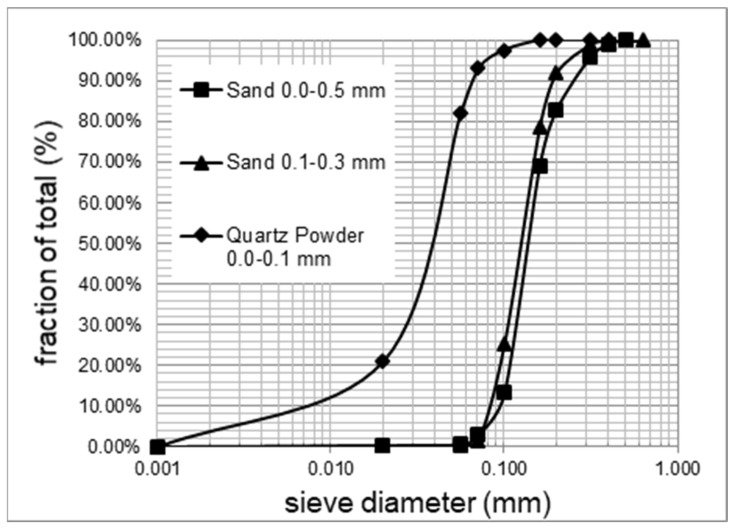
Particle size distribution of aggregates.

**Figure 2 materials-14-07558-f002:**
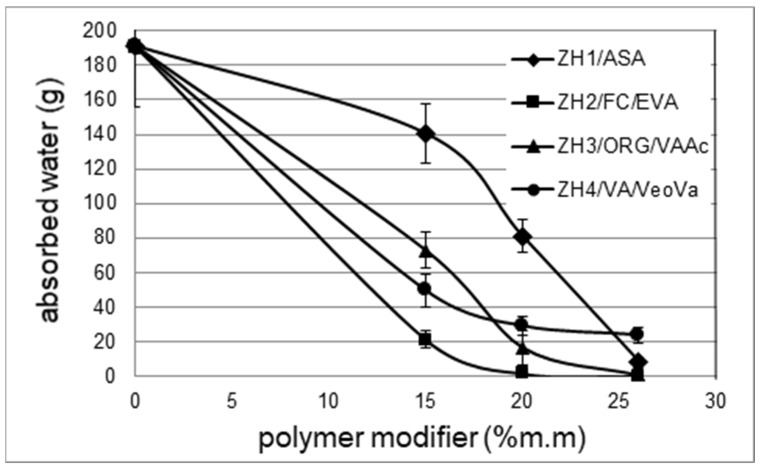
Dependence of water permeability expressed in absorbed water (g) on the mass fraction of polymer in the mortar.

**Figure 3 materials-14-07558-f003:**
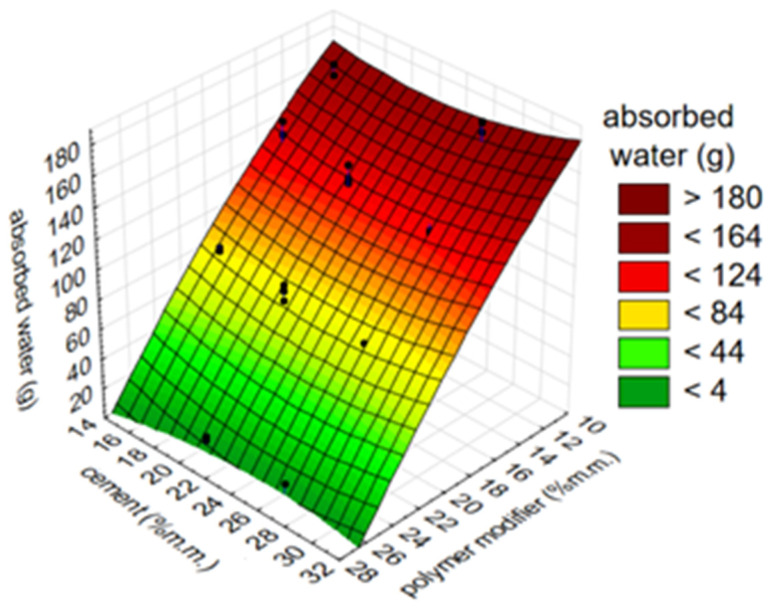
The effect of the simultaneous change in the amount of the ZH1/ASA polymer modifier and cement on the water permeability expressed in (g) of the absorbed water.

**Figure 4 materials-14-07558-f004:**
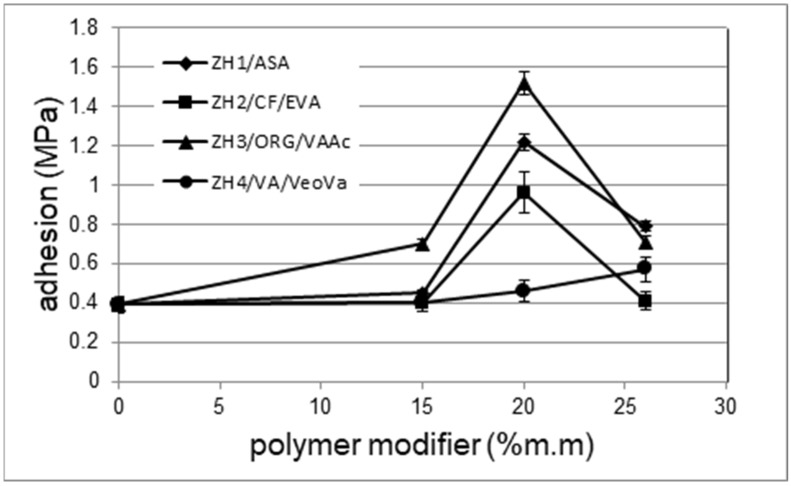
The relationship between the adhesion of the mortar to the concrete substrate depending on the proportion of the polymer modifier.

**Figure 5 materials-14-07558-f005:**
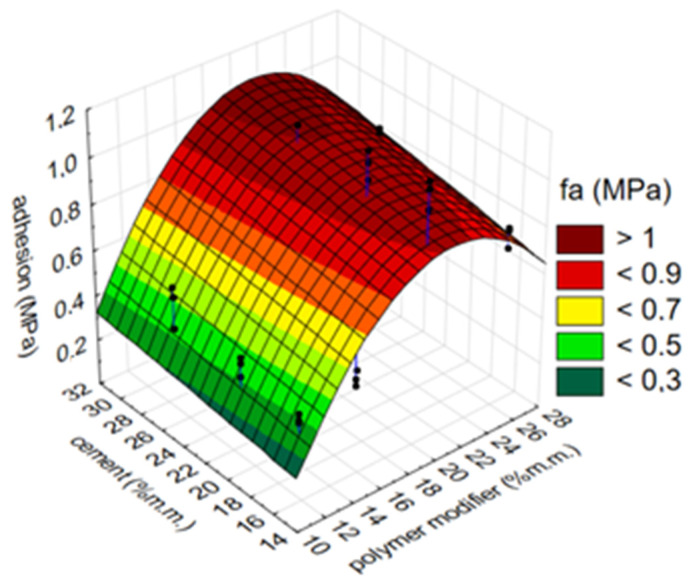
Influence of simultaneous change in the amount of ZH1/ASA polymer modifier and of cement on mortar adhesion.

**Figure 6 materials-14-07558-f006:**
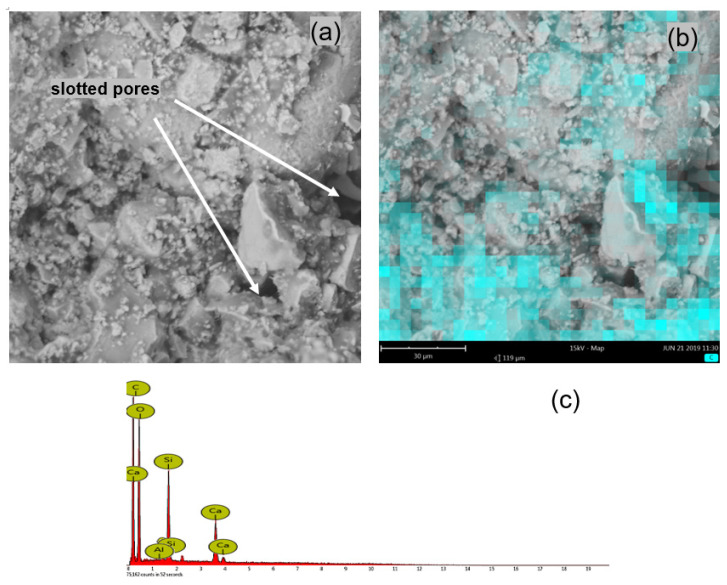
SEM photo of the mortar fracture: (**a**) modified with ZH1/ASA in the amount of 15% m/m, (**b**) SEM with marked areas of the polymer modifier occurrence obtained by mapping C atoms on the surface of the polymer–cement mortar fracture, (**c**) EDS analysis of the scanned fracture surface.

**Figure 7 materials-14-07558-f007:**
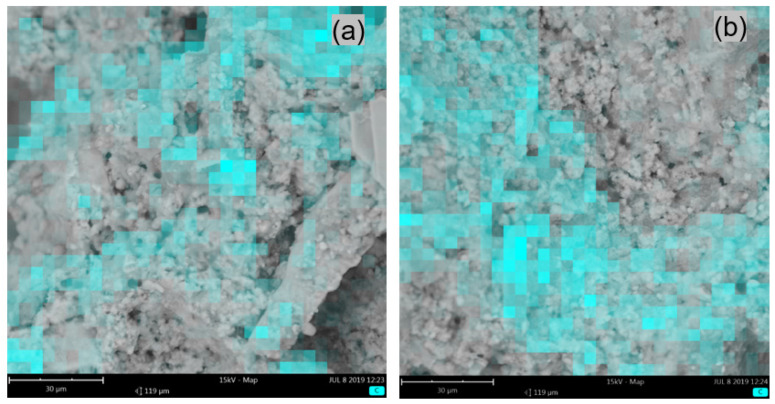
SEM photo of the mortar samples fractures with marked areas of C atoms occurrence obtained by mapping the carbon atoms of mortars modified with ZH3/ORG/VAAc, containing: (**a**) 15% m/m of the modifier, (**b**) 20% m/m the modifier.

**Figure 8 materials-14-07558-f008:**
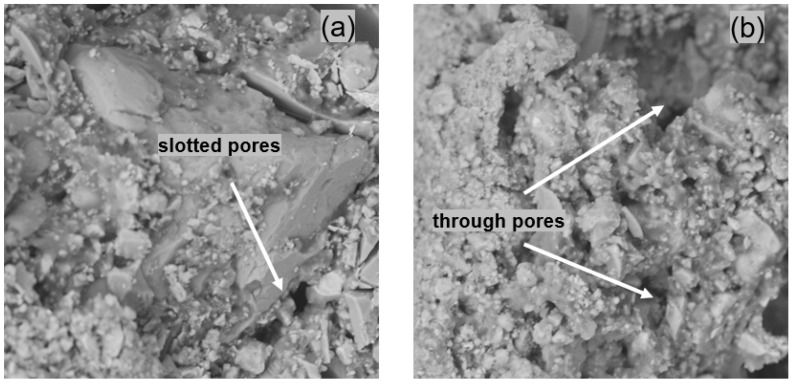
View of the fracture microstructure of the mortar modified with ZH1/ASA: (**a**) taken from the inner part of the mortar with 20% m/m of the modifier with slotted pores marked; (**b**) view of the fracture with circular pores in the mortar inter-grain space marked.

**Figure 9 materials-14-07558-f009:**
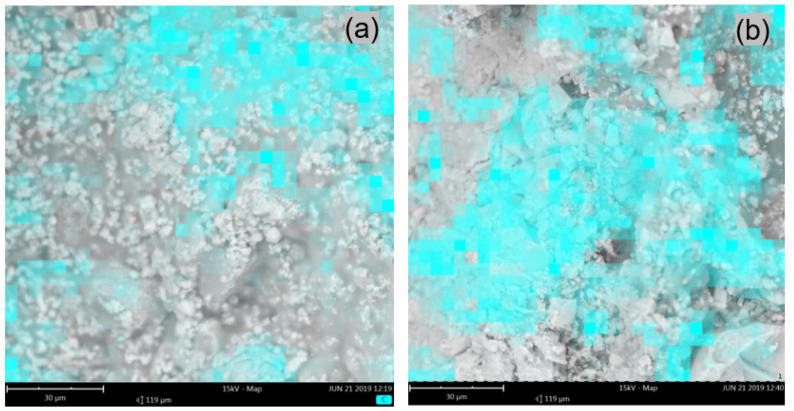
SEM photos of the mortar sample with the areas of carbon atoms marked, obtained by mapping the carbon atoms of the mortar modified with ZH2/CF/EVA in the amount of 26% m/m (**a**) on the surface of the mortar and (**b**) in the mortar–substrate interface.

**Figure 10 materials-14-07558-f010:**
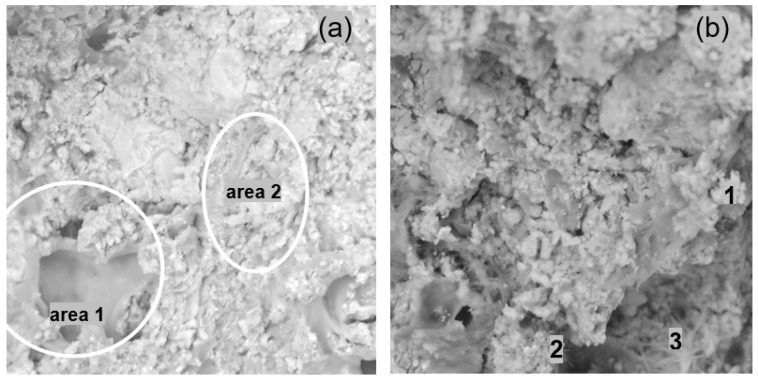
View of the surface of the mortar modified with ZH4/VA/VeoVa in the amount of 20% m/m, where: (**a**) area 1—a place with a well-formed polymer film, area 2—area containing a mineral binder; (**b**) view of the mortar fracture where: 1, 2—hydrates of irregular shape with a large number of isometric and flattened particles, 3—fibrous-shaped C-S-H phase.

**Figure 11 materials-14-07558-f011:**
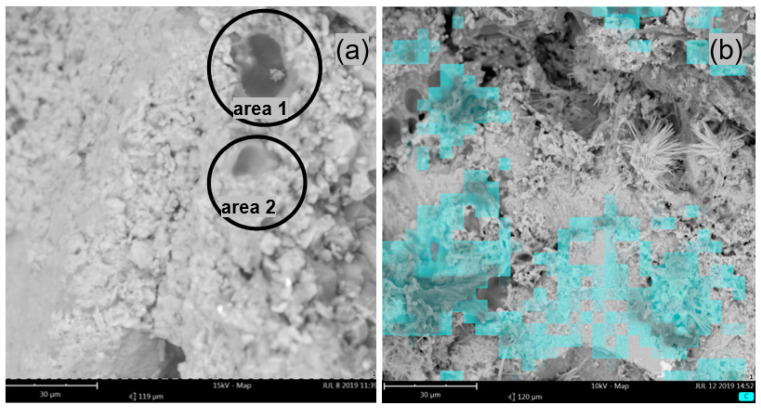
SEM photos of the mortar fracture taken from the inner part of the mortar sample modified with ZH4/VA/VeoVa in the amount of 20% m/m (**a**), where area 1 and area 2—pores with a circular cross-section filled with hardened polymer, (**b**) fragment with marked areas of polymer modifier occurrence obtained by mapping C atoms.

**Table 1 materials-14-07558-t001:** Chemical and mineral composition of CEM I 42.5R Portland cement.

Chemical Composition [% by Weight]										Mineral Composition [% by Weight]			
Loss on ignition	SiO_2_	AL_2_O_3_	Fe_2_O_3_	CaO	MgO	SO_3_	Cl^-^	Na_2_O	K_2_O	C_3_S	C_2_S	C_3_A	C_4_AF
2.24	20.6	5.0	2.6	64.2	1.4	2.9	0.05	0.15	0.78	67.0	12.8	9.1	7.6

**Table 2 materials-14-07558-t002:** Physical and mechanical properties of the cement used.

Property	Unit	Obtained Result
Specific surface area	cm^2^/g	3800
Constancy of volume	mm	0.2
Initial setting time	minutes	202
Compressive strength after:		
2 days	MPa	28.8
28 days	MPa	58.2

**Table 3 materials-14-07558-t003:** Composition of the test mortar.

Component	Portion (% m/m)
CEM I 42.5 R Portland cement	15.00
CL 90-S Ca(OH)_2_ hydrated lime	3.00
Polymer modifier	15.0%; 20.0%; 26.0%
Quartz powder 0.0–0.1	1.00
Quartz sand 0.1–0.3	4.03
Quartz sand 0.0–0.5	55.90
Modified Methyl Hydroxy Propyl Cellulose	0.10
Starch ether 15 mPas	0.03
Cellulose fibre (length 300–700 μm)	0.34
Polypropylene fibre FPE	0.60
Total dry weight	100
Water	28.00
Water/cement (w/c) ratio	1.87

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
