# Peer review of "Influence of Polymer Modifiers on Selected Properties and Microstructure of Cement Waterproofing Mortars"

_materials, 2021, doi:10.3390/ma14247558_

Round 1
Reviewer 1 Report
The paper presents a study on the effect of four polymers in concentrations of 15, 20 and 26 % by mass of the dry components of mortar, on the water permeability and adhesion to the concrete substrate. The results do not support some of the conclusions given by the authors. Sample preparation for SEM analysis were just broken pieces that do not permit porosity and pore size analysis to make an objective comparison of the distinct polymers used. I would recommend rejecting the paper.
I have the following specific comments:
- Was there any reason to select the 26 % content of the polymers, since we would expect 25 %, according to the lower concentrations of 15 % and 20 %.
- The water/cement ratio seems too high (1.87 or 187 %). Please check the way of expressing correctly this parameter.
- Please use the common terms for description of the cement properties. For instance, Start of setting time is normally known as initial setting time. What is the final setting time? What is constancy of volume?
- The quality of the graphs, for instance Fig.2, must be improved. Please indicate the error bars. Also, it seems to be incorrect the term watertightness (in grams) for the Y axis. It is the absorbed water.
- In page 5, line 172, the statement “A further increase in the amount of the modifier had no significant effect on water permeability”, is incorrect, since at 20 % the absorption in near zero, lower then the previous at 20 g. Of course, if the standard deviation is low, it perhaps can be stated that there is no significant effect. However, there is no standard deviation presented.
- In page 6, line 189, the term adsorbed should be absorbed.
- Page 10, figure 8. The authors introduce the terms slotted pores and through pores. This description is not usual. What would be the difference with capillary pores?. In broken surfaces it is hard to see the actual porosity and size of pores. The samples to observe and quantify the porosity should have a flat and polished surface. This would allow image analysis to obtain amount and pore size distribution, so an objective comparison is done for the different polymers used in the experiment.
- The maps of C distribution would be more reliable if obtained from flat polished samples, since they will be less affected by different depths present in a broken surface. As the figures show, there is not an even distribution of the polymer in the mortar. In addition, the pixel resolution of the maps is very different from the resolution of the secondary electrons images.
- In the discussion section, the authors relate the increased watertightness of the mortar containing polymers with the gradual filling of the micropores with the polymer. Why the micropores would be affected instead of the meso or macro pores that are more related to the permeability? What would be evidence of this filling?
- The statement in page 13, lines 346 and 347, related to the even dispersion of the polymer within the mortar and also that there is a good adhesion to the aggregates, how this latter is demonstrated by the results?. The maps of C do not show an even dispersion of the polymer.
Author Response
Thank you for taking the time to review our article. In accordance with your suggestions, we have made changes to its content. Changed fragments are marked in green. Please find below our responses to your comments:
- The amount of polymer (26%) was dictated by technological considerations (because the product was put into production)
- The high value of the w / c ratio was due to the large amount of other ingredients showing high water retention capacity, incl. cellulose fibers
- The cement specification was made in accordance with the PN-EN 191. This standard does not require specifying the final setting time, therefore the info has not been specified
- The term "watertightness" has been replaced with the term "absorbed water"
- Standard deviation has been implemented to the graphs in the article
- The error has been corrected
- and 8. Thank you for the valuable comment regarding the possibility of polishing the samples in order to determine the pore size distribution. The aim of the reviewed article was not to determine the pore size distribution, but we focused more on the polymer distribution in terms of mortar homogeneity. The distribution of the polymer in the samples was determined by analysis of carbon atoms. In our opinion, polishing the samples could lead to accelerated carbonation of cement hydration products, which we wanted to avoid. In order to avoid carbonation, the samples were kept in a vacuum until the test.
- The above comments also apply to point 8.
- Right remark, the correction has been implemented. The sample data tested showed a wide spectrum of porosity, from micro, through meso, to macropores. As the polymer concentration increased, they were gradually filled with the polymer.
- The statement about the uniform distribution of the polymer was based on a wide range of studies, however, due to the large number of photos taken and analyzed, we were not able to include all of them. Very fair remark that the photo presented in the article is a bit unfortunate. In the final version, the photo was changed.

Reviewer 2 Report
This is an interesting work and timely work. Some comments:
- The abstract is easy to follow.
- The introduction reviewed the effect of different polymer additives. However, the experimental methods are not mentioned, please revise them.
- Figure 1: the control sieving size should be marked.
- Line 133: how the authors decide 15%, 20%, and 26%?
- Figure 6c: looks like this figure is not complete. Please double check.
- The analysis and conclusions are reliable.
Author Response
Thank you for taking the time to review our article. In accordance with your suggestions, we have made changes to its content. Please find below our responses to some of your comments:
4. The amounts charged are based on previously conducted exploratory research. It was decided to use these amounts of polymer because they show the greatest variability in watertightness. With the maximum proportion of the tested polymer (26%), the amount of absorbed water was close to zero and further analysis would not make sense. The maximum amount of polymer (26%) was dictated by technological considerations.
5. Only selected essential elements were analyzed.
Reviewer 3 Report
The presented article has sufficient scientific significance and can be recommended for publication.
But there are some comments that could improve the quality of the article:
1) Fig. 5 would need to be explained. Why does not the adhesion decrease by reducing the amount of Portland cement? There is no explanation for this fact.
2) the porosity values ​​are very important for the "water permeability" indicator. This data is missing in the article.
3) the form and quantity of CSH cannot be judged only by the data of electron microscopy. It is necessary to apply a set of studies: XRF, DTA, IR spectroscopy.
Author Response
Thank you very much for the thorough analysis of the content of the article and the submitted comments. The comments are very important to us and will inspire further research in this area.
1. Note included - additional explanations are provided in the text on page 8 (marked with green). In the part 4 (page 16) of the article there is an explanation of this mechanism.
2. The comment is right and important. In our research, we used water permeability to assess tightness. This feature is obviously related to the porosity structure. Certainly, a thorough understanding of the porosity structure would allow for a more detailed analysis of the discussed issue. We will take this suggestion into account in our next research work.
3. The comment is right and important. We made a suggestion regarding the form and amount of the CSH phase based on the available data in the literature on the habit of the CSH phase, which is the main product of CEM I Portland cement hydration (approx. 70%) [1,2]. The given test methods are known to us and will be used in further research.
1.C. Famy, AR. Brough, H.F.W. Taylor:The C-S-H gel of Portland cement mortars: Part I. The interpretation of energy-dispersive X-ray microanalyses from scanning electron microscopy, with some observations on C-S-H, AFm and AFt Cement Conrete Research, vol.33, 2003, pp. 1389-1398,
2. W. NocuÅ„ Wczelik; Structure and properties of hydrated calcium silicates, Polish Ceramic Bulletin 21, Polish Academy of Science, Kraków 1992, (in Polish)
Once again, I would like to thank for valuable comments that will increase the value of the article.

Round 2
Reviewer 1 Report
In their response, the authors say they changed a figure but I did not find it and they do not say which figure was changed. The figures are exactly the same as in the previous version.
The reviewer still thinking that a qualitative microstructure evaluation from mapping the carbon in broken samples is not a good enough because of the different depths of each point in the sample that will interact differently with the electron beam. Also, measurements of porosity in a broken sample can not be correctly assessed. I would suggest the authors to try to polish the samples and obtain back scattered electron images. Then, these images may be analyzed with an image analysis software such as Image-J to extract porosity and pore size distribution. Also the carbon mapping would be more reliable in polished samples or even in samples just cut with a low speed saw that leaves a good surface.
Author Response
Thank you for the valuable comments.
We are very sorry, in the previously shared version, we did not put the changed figures by mistake. In the newest version of the article, according to your suggestion we have changed figures 2, 3, 4, 5, 9b and 11b.
All changes to the text have been marked with green color.
The test samples were prepared using a low speed saw. The article has been supplemented with this information. On the other hand, in future research we will take into account both proposed methodologies for preparing test samples (cutting, grinding) and compare the obtained results. We are not able to do this research in the near future in order to make appropriate additions to the assessed work.
